# Burden of Mental Illness among Primary HIV Discharges: A Retrospective Analysis of Inpatient Data

**DOI:** 10.3390/healthcare10050804

**Published:** 2022-04-26

**Authors:** Robert M. Avina, Jim E. Banta, Ronald Mataya, Benjamin J. Becerra, Monideepa B. Becerra

**Affiliations:** 1School of Public Health, Loma Linda University, Loma Linda, CA 92354, USA; jbanta@llu.edu (J.E.B.); rmataya@llu.edu (R.M.); 2Center for Health Equity, Department of Information and Decision Sciences, California State University, San Bernardino, CA 92407, USA; benjamin.becerra@csusb.edu; 3Center for Health Equity, Department of Health Science and Human Ecology, California State University, San Bernardino, CA 92407, USA; mbecerra@csusb.edu

**Keywords:** HIV, mental health, length of stay, post-discharge, routine disposition, mental illness

## Abstract

Background: Empirical evidence demonstrates the substantial burden of mental illness among people living with HIV and AIDS (PLWHA). Current literature also notes the co-morbidity of these two illnesses and its impact on quality of life and mortality. However, little evidence exists on patient outcomes, such as hospital length of stay or post-discharge status. Methods: A retrospective analysis of National Inpatient Sample data was conducted. The study population was defined as discharges having a primary diagnosis of HIV based on International Classification of Disease, 10th Revision, Clinical Modification (ICD-10-CM) codes in primary diagnosis field. Clinical Classification Software (CCS) codes are used to identify comorbid mental illness. Length of stay was defined as number of days between hospital admission and discharge. Disposition (or post-discharge status) was defined as routine versus not routine. Patient and hospital characteristics were used as control variables. All regression analyses were survey-weighted and adjusted for control variables. Results: The weighted population size (N) for this study was 26,055 (n = 5211). Among primary HIV discharges, presence of any mental illness as a secondary discharge was associated with 12% higher LOS, when compared to a lack of such comorbidity (incidence rate ratio [IRR] = 1.12, 95% confidence interval [CI] = 1.05, 1.22, *p* < 0.01). Likewise, among primary HIV discharges, those with mental illness had a 21% lower routine disposition, when compared to those without any mental illness (OR = 0.79, 95% CI = 0.68, 0.91, *p* < 0.001). Conclusion: Our results highlight the need for improved mental health screening and coordinated care to reduce the burden of mental illness among HIV discharges.

## 1. Introduction

The Human Immunodeficiency Virus (HIV) is classified into two strands or variants, HIV-1 and HIV-2. Both strands share common similarities such as modes of transmission, gene arrangement, and intracellular replication pathways. However, without proper treatment, both strands can progress to Acquired Immune Deficiency Syndrome (AIDS) [1]. HIV-2 has been characterized by lower transmissibility, as well as reduced likelihood of progressing to AIDS. In addition, HIV-2 is mostly confined to West Africa, whereas HIV-1 is identified worldwide [1]. HIV continues to be a major global public health issue impacting an estimated 37.9 million people globally, including 1.7 million children [2]. In 2018, the United States reported 1.1 million people living with HIV, with an estimated 37,832 new case [3]. In addition, from 2010 to 2017, the United States identified 325,502 HIV-1 and 198 HIV-2 infections [4].

People living with HIV/AIDS (PLWHA) also face a plethora of social and economic barriers such as lack of support systems and loss of work [5]. Currently, the literature highlights that PLWHA may have poor quality of life (QOL). QOL consists of a person’s physical, social, and mental wellbeing including happiness and satisfaction with their life. The World Health Organization (WHO) defines QOL as an individual’s perception of their position in life. In addition, PLWHA are more likely to experience social avoidance that is associated with greater health related stress [6]. Furthermore, cost-effective analysis tools continue to be an effective method in identifying the average cost for a person with an HIV new infection, as well as the cost of lifetime HIV treatment. To illustrate, the Centers for Disease Control and Prevention (CDC) estimated that a person with a new HIV diagnosis has an estimated cost of USD 41,667, while the cost of an HIV lifetime diagnosis is estimate at USD 379,668 [7]. In addition, the Health Resources and Services Administration (HRSA) identified that the monthly breakdown for treating HIV ranges between USD 1854 and USD 4545 a month, which is much higher compared to that for an uninfected person, which ranges between USD 73 and USD 628 a month [8]. Undoubtedly, HIV continues to impact communities throughout the United States, causing stress and burden at the socioeconomic level, which impacts a person’s quality of life and increases overall healthcare costs.

Studies have highlighted that mental illness can further worsen the burden of HIV. In the United States, mental illnesses affect nearly one in five adults, which include different conditions that vary based on the degree of severity. In 2017, the United States reported that 46.6 million people are living with a mental illness ranging between moderate to severe [9]. Any mental illness (AMI) is present in 18.9% of all-American adults and prevalence appears to be higher among women by (22.3%) when compared to men (15.1%). In addition, prevalence of AMI was highest among individuals with two or more races (28.6%), while White adults (20.4%) and Asians (14.5%) were lowest [9]. An additional study identified that higher levels of income inequality are directly associated with a higher prevalence of mental illness. For example, individuals that experience socioeconomic disadvantage such as unemployment, low income, poverty, debt, and poor housing are associated with poorer mental health [10]. In addition, studies have also identified that a mental illness impacts a person’s QOL especially when their mental illness is severe, making them more likely to experience a range of chronic physical conditions. On the other hand, presence of mental and physical illness can also reduce and/or diminish a person’s QOL, which can lead to longer duration of illnesses and poorer health outcome. In addition, mental illnesses are an economic cost to society since there is reduced productivity and an increase in health-related services overall [11].

Mental illness has been shown to worsen length of stay (LOS) for various hospital visits due to other medical conditions. LOS is a metric that is based on the number of days that a patient stayed in an inpatient facility during a single episode of hospitalization [12]. It is a valuable metric utilized for healthcare providers and patients and is influenced by various factors such as the ability to identify the severity of illnesses, as well as healthcare resource utilization [13]. For example, LOS can be utilized as an indicator of treatment [14], as well as an indicator of assessing hospital efficiency [15]. One study found that prolonged LOS is associated with increased mortality and other negative health outcomes [16]. Other studies have identified that the reduction in the number of inpatient days leads to decreased risk of infection, medication side-effects, improvement of quality of treatment, and increased hospital profit with more efficient bed management [17].

Cumulatively, such literature highlights that presence of mental illness among various chronic illnesses can worsen outcomes. However, the burden of mental illness on patients and hospital outcomes of HIV discharges remains to be evaluated. This present study aims to address such a gap in the literature and evaluate the healthcare burden of mental illness among PLWHA with emphasis on whether presence of mental illness among those with primary HIV diagnosis increases hospital LOS and worsens post-discharge status.

## 2. Methods

The study was conducted by using a retrospective case–control analysis of the National Inpatient Sample (NIS). NIS uses a complex survey design, thus making results generalizable to the U.S. It also includes discharge data from all states that participate in Agency for Healthcare Research and Quality (AHRQ)-sponsored Healthcare Cost and Utilization Project (HCUP). It is the largest publicly available all-payer inpatient database in the U.S. [18]. Furthermore, NIS is reflective of a 20% stratified sample of all community hospitals, defined as non-federal, short-term, general, and other specialty hospitals, though it excludes short-term rehabilitation hospitals (starting with 1998 data), long-term non-acute care hospitals, psychiatric hospitals, and alcoholism/chemical dependency treatment centers. Finally, NIS further divides all hospitals into five strata based on ownership, setting, bed size, teaching status, and geographic location. Since 1998, NIS data continues to be released on a yearly basis [18].

## 3. Study Variables

NIS provides a primary discharge code for each patient and up to 24 secondary diagnoses codes. In this study, we used primary discharge code of HIV, identified using International Classification of Disease, 10th Revision, Clinical Modification (ICD-10-CM) codes B20, B97.35, and Z21, to select our study population. Next, we used NIS provided secondary diagnoses codes to identify our exposure variable of mental illness. We used Clinical Classifications Software (CCS) codes for ICD-10-CM for mood disorders (657), personality disorders (658), schizophrenia (659), suicide and intentional self-inflicted injury (662), impulse control disorders NEC (656), anxiety disorders (651), and adjustment disorders (650) for identifying mental illness [19]. The outcome variable for this study was LOS (days) in the hospital.

The secondary analysis of interest was to evaluate of presence of mental illness worsened patient disposition (routine versus not routine), indicating post-discharge status. Routine disposition is defined as discharged to home, while not routine is defined as non-ideal, such as other care facility, death, etc. This database only included those aged 18 years or older.

Covariates included were patient and hospital characteristics provided by NIS [18]. Patient characteristics were: age (18–34 years, 35–49 years, 50–64 years, 65 years or more), sex (male, female), race/ethnicity (White, Black, Hispanic, Other), neighborhood annual income quartile (USD 1–43,999, USD 44,000–55,999, USD 56,000–73,999, USD 74,000 or more), insurance payer type (Medicare, Medicaid, Private including HMO, Other), Elixhauser comorbidities from Elixhauser Comorbidity Software, Version 3.7 [20], and alcohol-related or substance-related disorders (CCS Codes 660,661). Hospital characteristics included: hospital control/ownership (private-investor-owned, private-non-profit, governmental-non-federal), bed size in tertiles (small, medium, large), hospital setting (rural, urban teaching, urban non-teaching), census region (Northeast, Midwest, South, and West).

## 4. Data Analyses

The first step in data analysis was to conduct descriptive statistics using survey weighted population estimates and percentages. Next, we ran bivariate analyses and obtained the prevalence of mental illness by patient and hospital characteristics. Statistical significance was assessed with a Chi-square test using design-based F values. To evaluate the role of mental illness on length of stay, multivariable negative binomial regression analysis was conducted with the aforementioned covariates. To assess patient disposition outcome (binary data), multivariable logistic regression analysis was used. All statistical analyses utilized appropriate survey weights and were conducted using SAS 9.4 (SAS Institute, Inc., Cary, NC, USA), while survey-weighted negative binomial regression was conducted using STATA 14 (STATA Corp., College Station, TX, USA). All analyses used a level of significance of 0.05.

## 5. Results

Table 1 displays the characteristics of the study sample. The weighted population size (N) for this study was 26,055 (n = 5211). Prevalence of mental illness in the study sample was 29.15% and 43.75% reported alcohol-related or substance-related disorders. Furthermore, a higher proportion of study sample characteristics were: aged 35–49 years (36.68%), males (70.22%), Blacks (54.46%), an income of USD 1–43,999 (52.03%), Medicaid (43.65%), comorbidity of fluid and electrolyte disorders (43.91%). In addition, higher proportion of hospital characteristics were: large bed sizes (58.16%), private, non-profit (67.93%), urban teaching (81.15%), and located in southern regions (52.54%).

In the overall HIV discharge population, LOS was 9.52 days, while 10 days among HIV discharges with mental illness and 9 days among discharges without mental illness.

Table 2 illustrates the results of bivariate analysis on the association of prevalence of mental illness and patient/hospital characteristics. A significant association was found between presence of mental illness and several of such characteristics. A significantly higher prevalence of mental illness was identified among ages 50–64 years (30.28%, *p* < 0.05), females (34.95%, *p* < 0.001), White (38.29%, *p* < 0.001), Medicare (35.80%, *p* < 0.001), urban teaching hospital status (30.43%, *p* < 0.001), and Midwest region (34.02%, *p* < 0.01). Additionally, mental illness was significantly associated with alcohol-related or substance-related disorders, as well as several comorbidities, as further detailed in Table 2.

Table 3 demonstrates the results of survey-weighted multivariable negative binomial regression. Among primary HIV discharges, those with secondary mental illness had a 12% higher LOS, when compared to those without any mental illness (incidence rate ratio [IRR] = 1.12, 95% confidence interval (CI) = 1.05, 1.22, *p* < 0.01).

On the other hand, 20% lower LOS (IRR = 0.80, 95% CI 0.75,0.86, *p* < 0.001) was identified among those with alcohol-related or substance-related disorders. Elixhauser co-morbidities associated with an increase in LOS were: those with congestive heart failure, pulmonary, peripheral vascular disease, paralysis, other neurological disorders, peptic ulcer disease and bleeding, lymphoma, coagulopathy, weight loss, fluid and electrolyte disorders, and deficiency anemias. Pulmonary circulation disease was associated with lower LOS

Additional results show that LOS among Blacks when compared to Whites was 10% higher (IRR = 1.12, 95% CI = 1.01, 1.20, *p* < 0.05). LOS was 28% higher (IRR = 1.28, 95% CI = 1.17, 1.39, *p* < 0.001) among those that had Medicaid, 16% higher (IRR = 1.16, 95% CI 1.05, 1.28, *p* < 0.001) among those with private including HMO, and 27% higher (IRR = 1.27, 95% CI 1.09, 1.33, *p* < 0.001) among those with other insurance type when comparing to those that had Medicare.

Furthermore, when comparing bed sizes, medium hospital bed size had a 21% higher LOS (IRR = 1.21, 95% CI 1.09, 1.33, *p* < 0.001), while large hospital bed size was 29% higher LOS (IRR = 1.29, 95% CI 1.18, 1.41, *p* < 0.001), when compared to small hospital bed size. Hospitals that were private, non-profit had a 12% lower LOS (IRR = 0.88, 95% CI 0.80, 0.98, *p* < 0.05) when compared to government, nonfederal hospitals. In addition, urban teaching hospitals had a 30% higher LOS (IRR = 1.30, 95% CI 1.08, 1.56, *p* < 0.01) when compared to rural hospitals. Hospitals that were in the Midwest had a 24% lower LOS (IRR = 0.76, 95% CI 0.67, 0.86, *p* < 0.001) compared to hospitals in the Northeast.

For our sub-analysis focused on patient disposition, results show that the prevalence of routine disposition in the overall study population (HIV discharges) was 62.98%, while among HIV discharges with mental illness it was lower (58.31%) when compared to those without such mental illness (64.90%). Regression results demonstrate that among primary HIV discharges, those with secondary mental illness had a 21% lower routine disposition, when compared to those without any mental illness (OR = 0.79, 95% CI = 0.68, 0.91, *p* < 0.001).

## 6. Discussion

This study aimed to evaluate the role of mental illness on LOS among patients with primary HIV discharges. A major finding of the study was that the presence of mental illness increased LOS among such HIV discharges by 12% and decreased the likelihood of routine post-discharge status by 21%. This was consistent with previous literature among chronic illnesses, though the results provide novel contribution to the field on HIV/AIDS-mental illness nexus.

For example, Wancata et al. [21] reported that patients with several somatic diagnoses (in which physical symptoms can cause emotional distress) have longer LOS when compared to patients that were only suffering from one physical illness. Likewise, newer studies noted similar trends. Likewise, Koopmans et al. [22], in evaluating 23 studies in a comprehensive review, identified partial association between a psychiatric comorbidity and LOS. Bressi and colleagues [23] found that patients with psychiatric conditions universally stayed in the hospital longer than patients without such conditions. LOS was also reported to be higher among heart failure patients with psychiatric comorbidities, compared to those without [24]. Siddiqui et al. [25] found that among 16,989 patients hospitalized with a chronic medical condition and with mental illness as a comorbidity experienced longer LOS, as compared to those without the mental illness comorbidity. In addition, such patients also incurred higher bed days use when compared to those without a mental illness. As such, the authors hypothesize that such increased LOS was a further reflection on lower quality of life and care experience. Jansen et al. [26], in a systematic review, also identified that hospital patients with a medical-psychiatric comorbidities led to an increase in LOS when compared to patients without such comorbidity. While such literature consistently highlights the burden of mental illness among patient hospitalization, primarily with chronic conditions, the results of this study further provide applicable outcomes among HIV patients, thus providing a unique contribution to the field of literature on LOS.

Such results also provide scopes of public health and healthcare practice. For example, local and community-based preventative and treatment strategies that integrate feedback can be valuable for mental health professionals. Researchers suggest that mental health professionals can provide several supporting roles, such as ensuring adherence to treatment, address psychosocial issues, among others, to ensure healthy outcomes for HIV patients [27]. This also provides an opportunity for integrated care such that, mental and physical healthcare services can be better coordinated for improved benefits for patients [28].

Additional results of this study warrant further discussion. While there remains limited literature on racial disparity on LOS, a study by Pines et al. [29] noted that Blacks were more likely to have longer emergency department LOS, as compared to Whites. The authors noted that this increase in LOS may be attributable to a plethora of coordinated care needs, which is reflective of quality of care. On the other hand, Moss [30] noted that Blacks have shorter LOS, though the study was limited to women only. In addition, Moss notes that shorter time is a reflection of lower quality of care among Blacks, since they are not given the same level of time duration and services. While both these papers provide opposing results, the interpretation remains the same, which is, quality of care. In addition, evidence supports that LOS is related to quality of care, though this remains to be further evaluated. Given that non-routine disposition is indicative of less than ideal outcomes, such as transfer or death, there remains a critical importance of improving long-term quality of care of PLWHA by addressing co-morbid conditions.

The results of this study expand what is noted by Pines et al. [9] to that of inpatient care and suggest that the increased in LOS among Blacks maybe reflective of slower processes of care and thus indicate reduced time and quality. On the other hand, it is feasible that the increased LOS could be due to an accumulation of additional illnesses. The current data further notes [31] that Blacks have higher health disparities, often due to systemic structural barriers as one of many disadvantages faced, which in turn could have attributed to additional care needs and thus longer LOS. While this paper does not address these factors, it provides the foundation for additional studies to address this racial disparity.

In addition, our results noted that LOS was higher among those with Medicaid, private including HMO, and those with other insurance type as compared to Medicare. Medicare is a federal health insurance program for those who are 65 years or older, people with disabilities, and/or people with end stage renal disease [32]. Over the years, Medicare’s fee-for-service system has provided incentives to reduce LOS; however this has only led to an increase in Medicare patients in skilled nursing facilities [33]. The intentions behind reducing LOS was to decrease hospital costs and improve patient health outcomes, but in return patient health outcomes have not improved in all areas [33]. While most recent literature is lacking, a historical study from 1993 examined the Medicare system and the affect it has on LOS. For instance, Kominski & Witsberger [34] examined the Medicare system between 1979 and 1987. The authors noted that in 1979 the LOS trend was decreasing; however, in 1987 the LOS trend was increasing, identifying two different trends. After further examination, the authors identified that LOS was decreasing for most procedures but was offset by an increased shift towards more complex procedures leading to a rise in LOS. Furthermore, patients that have Medicare reported higher LOS and Gornick [35] identified that inpatient hospital services increase significantly with age. Such historical literature could putatively explain the results noted in our study and further provides the foundation for additional studies to address the impact on LOS and insurance payer type.

While there remains limited literature on the role of teaching hospital status on LOS, a study by Hyder et al. [36] noted that patients at teaching hospitals are more likely to experience higher LOS, which remains consistent with our findings. When comparing urban hospitals to rural hospitals, rural hospitals are much smaller than urban hospitals due to the average bed sizes within the facility [36]. While more recent literature was lacking, a historical study from 1986 showed that urban hospitals have around 76 to 252 beds per hospital, while the average LOS can vary based between states [37]. For example, urban hospitals within states such as New York have higher LOS, whereas rural hospitals had significantly lower LOS within the state. In addition, teaching hospitals within an urban location are more likely to have specialized procedures, which in return leads to an increase in LOS while rural hospitals might not. Furthermore, urban teaching hospitals have higher bed sizes and are located in a highly densely populated areas, which are more likely to experience higher patient volumes when comparing to rural hospitals in low densely populated areas [37]. Although this paper was not centered on addressing these factors, it provides initial elements needed for future studies to address the differences between urban teaching hospitals and rural teaching hospitals.

The results of this study should be interpreted in the context of its limitations. As an administrative dataset there was a lack of socio-demographic such as education, marital status, and living arrangement. Additionally, HIV severity indicators such as CD4 count were not included in the dataset and thus could not be assessed in our analysis. Since discharges have been de-identified, readmissions cannot be determined. Lastly, there may be hospital-level variation in reporting of secondary diagnoses of mental illness. Notwithstanding such limitation, our study has several strengths and thus contributions to the field. Unlike majority of studies that were limited to specific hospitals, our study utilizes nationally representative data, which allows for generalizability of the results due to increased external validity. For example, NIS data includes the uninsured, public, and private payers, unlike other databases, such as Medicare claims. Furthermore, our study adds towards the body of literature by providing a comprehensive assessment by using the largest inpatient database in the United States. In addition, while much of the literature highlights the co-morbid conditions of HIV/AIDS and mental illness, few have evaluated the outcome of such syndemics. Our results show that patients with such comorbidities have increased healthcare utilization and thus call for integrated healthcare where mental illness screening and early prevention are critical part of primary care encounters Remien et al., 2019).

## 7. Conclusions

Our study results expand the current literature on the HIV/AIDS–mental illness nexus. While past literature has established the co-morbid conditions of mental illness and HIV/AIDS, our study expands the empirical evidence to address the burden of such co-morbidity on such patients’ hospital outcomes. Our results highlight that PLWHA who also have co-morbid mental illness are likely to stay in the hospital longer than those without. Results noted that the burden of mental illness–HIV/AIDS comorbidity was comparable to PLWHA with congestive heart failure (14% increase in LOS), and remained significant when compared to other leading causes or morbidity, such as chronic pulmonary disease, diabetes, anemia, etc. Further studies are needed to address the putative foundations of the impact on LOS, include need for care, potential delays, lack of early prevention, etc. Studies addressing the importance of improve quality of care, such as early diagnosis and prevention of mental illness are needed.

## Figures and Tables

**Table 1 healthcare-10-00804-t001:** Characteristics of primary HIV discharges in the United States.

Variables	Population Estimate (N)	Survey-Weighted Percent
**Presence of any Mental Illness**	7595	29.15
**Alcohol-related or Substance-related Disorders**	11,400	43.75
**Elixhausers co-morbidities**		
Congestive Heart Failure	2265	7.42
Valvular Disease	580	1.90
Pulmonary Circulation Disease	355	1.16
Peripheral Vascular Disease	325	1.06
Paralysis	940	3.08
Other Neurological Disorders	3155	10.33
Chronic Pulmonary Disease	5960	19.52
Diabetes w/o Chronic Complications	1240	4.06
Diabetes with Chronic Complications	1720	5.63
Hypothyroidism	830	2.72
Renal Failure	3825	12.53
Liver Disease	3170	10.40
Peptic Ulcer Disease Bleeding	215	0.70
Lymphoma	1500	4.91
Metastatic cancer	435	1.42
Solid Tumor without Metastasis	475	1.60
Rheumatoid Arthritis/Collagen Vascular Disease	265	0.87
Coagulopathy	3745	12.27
Obesity	1385	4.54
Weight Loss	7630	25.00
Fluid and Electrolyte Disorders	13,405	43.91
Chronic Blood Loss Anemia	200	0.66
Deficiency Anemias	9850	32.30
Hypertension	8430	27.61
**Demographics**		
**Age in Years**		
18–34 years	6215	23.90
35–49 years	9535	36.68
50–64 years	8720	33.54
65 years +	1525	5.86
**Sex**		
Male	18,290	70.22
Female	7755	29.77
**Race/Ethnicity**		
White	6320	24.79
Black	13,880	54.46
Hispanic	3860	15.14
Other	1425	5.59
**Income**		
USD 1–43,999	13,155	52.03
USD 44,000–55,999	5335	21.10
USD 56,000–73,999	4165	16.47
USD 74,000 or more	2625	10.38
**Health Insurance Type**		
Medicare	6565	25.25
Medicaid	11,350	43.65
Private including HMO	4585	17.63
Other	3500	13.46
**Hospital Characteristics**		
**Hospital Bed Size**		
Small	3975	15.25
Medium	6925	26.57
Large	15,155	58.16
**Hospital Control**		
Government, Nonfederal	5120	19.65
Private, Non-profit	17,700	67.93
Private, Investor-owned	3235	12.41
**Hospital Rural/Urban Status**		
Rural	740	2.84
Urban Non-teaching	4170	16.00
Urban Teaching	21,145	81.15
**Hospital Region**		
Northeast	5020	19.26
Midwest	3175	12.18
South	13,690	52.54
West	4170	16.00

**Table 2 healthcare-10-00804-t002:** Prevalence of any mental illness by patient and hospital characteristics of primary HIV discharges in the United States.

Variables	Survey-Weighted Prevalence (%)
**Elixhauser Co-morbidities**	
Congestive Heart Failure	29.58
Valvular Disease	67.24
Pulmonary Circulation Disease *	40.85
Peripheral Vascular Disease	23.08
Paralysis	32.45
Other Neurological Disorders ***	35.66
Chronic Pulmonary Disease ***	37.58
Diabetes without Chronic Complications	28.63
Diabetes with Chronic Complications	28.49
Hypothyroidism ***	44.58
Renal Failure **	25.10
Liver Disease ***	34.86
Peptic Ulcer Disease Bleeding *	16.28
Lymphoma	25.00
Metastatic cancer	25.29
Solid Tumor without Metastasis	33.68
Rheumatoid Arthritis/Collagen Vascular Disease	35.85
Coagulopathy	27.10
Obesity	33.57
Weight Loss	28.83
Fluid and Electrolyte Disorders **	27.45
Chronic Blood Loss Anemia **	10.00
Deficiency Anemias *	29.34
Hypertension *	31.26
**Alcohol-related or Substance-related Disorders *****	37.32
**Demographics**	
**Age in Years ***	
18–34 years	27.92
35–49 years	30.05
50–64 years	30.28
65 years or more	22.30
**Sex *****	
Male	26.68
Female	34.95
**Race/Ethnicity *****	
White	38.29
Black	26.51
Hispanic	25.26
Other	24.91
**Income**	
USD 1–43,999	29.34
USD 44,000–55,999	30.55
USD 56,000–73,999	29.17
USD 74,000 or more	24.76
**Health Insurance Type *****	
Medicare	35.80
Medicaid	29.78
Private including HMO	21.81
Other	24.57
**Hospital Characteristics**	
**Hospital Bed Size**	
Small	30.31
Medium	27.29
Large	29.69
**Hospital Control**	
Governmental, Nonfederal	28.13
Private, Non-profit	29.55
Private, Investor-own	28.59
**Hospital Rural/Urban Status *****	
Rural	26.35
Urban Non-teaching	23.14
Urban Teaching	30.43
**Hospital Region ****	
Northeast	32.37
Midwest	34.02
South	27.25
West	27.82

* *p* < 0.05, ** *p* < 0.01, *** *p* < 0.001.

**Table 3 healthcare-10-00804-t003:** Impact of secondary mental illness on hospital length of stay (LOS) among primary HIV discharges in the United States.

Variables	IRR (95% CI)
**Any Mental Illness ****	1.12 (1.05, 1.21)
**Alcohol-related or Substance-related Disorders *****	
Absence	Reference
Presence	0.80 (0.75, 0.86)
**Elixhauser Co-morbidities**	
No Co-morbidity	Reference
Congestive Heart Failure	1.14 (1.00, 1.30)
Valvular Disease	0.91 (0.75, 1.11)
Pulmonary Circulation Disease **	1.52 (1.17, 1.99)
Peripheral Vascular Disease *	1.37 (1.06, 1.76)
Paralysis ***	1.78 (1.45, 2.19)
Other Neurological Disorders ***	1.50 (1.33, 1.69)
Chronic Pulmonary Disease	0.98 (0.91, 1.06)
Diabetes without Chronic Complications	1.16 (0.88, 1.54)
Diabetes with Chronic Complications	1.07 (0.93, 1.23)
Hypothyroidism	1.07 (0.86, 1.32)
Renal Failure	1.08 (0.98, 1.20)
Liver Disease	1.07 (0.94, 1.22)
Peptic Ulcer Disease Bleeding *	1.35 (1.04, 1.77)
Lymphoma ***	1.25 (1.12, 1.39)
Metastatic cancer	1.04 (0.85, 1.29)
Solid Tumor without Metastasis	0.99 (0.84, 1.17)
Rheumatoid Arthritis/Collagen Vascular Disease	1.21 (0.92, 1.60)
Coagulopathy ***	1.24 (1.15, 1.34)
Obesity	1.10 (0.93, 1.30)
Weight Loss ***	1.40 (1.31, 1.50)
Fluid and Electrolyte Disorders ***	1.41 (1.31, 1.51)
Chronic Blood Loss Anemia	1.09 (0.82, 1.45)
Deficiency Anemias *	1.07 (1.00, 1.14)
Hypertension	1.01 (0.93, 1.10)
**Demographics**	
**Age in Years**	
18–34 years	Reference
35–49 years	1.01 (0.92, 1.10)
50–64 years	1.03 (0.95, 1.12)
65 years or more	1.09 (0.94, 1.26)
**Sex**	
Male	Reference
Female	0.95 (0.88, 1.03)
**Race/Ethnicity**	
White	Reference
Black *	1.10 (1.01, 1.20)
Hispanic	1.07 (0.96, 1.20)
Other	1.00 (0.87, 1.14)
**Income**	
USD 74,000 or more	Reference
USD 1–43,999	1.00 (0.89, 1.12)
USD 44,000–55,999	1.05 (0.93, 1.19)
USD 56,000–73,999	1.06 (0.94, 1.20)
**Health Insurance Type**	
Medicare	Reference
Medicaid ***	1.28 (1.17, 1.39)
Private including HMO **	1.16 (1.05, 1.28)
Other **	1.27 (1.12, 1.44)
**Hospital Characteristics**	
**Hospital Bed Size**	
Small	Reference
Medium ***	1.21 (1.09, 1.33)
Large ***	1.29 (1.18, 1.41)
**Hospital Control**	
Governmental, Nonfederal	Reference
Private, Non-profit *	0.88 (0.80, 0.98)
Private, Investor-own	0.92 (0.80, 1.05)
**Hospital Rural/Urban Status**	
Rural	Reference
Urban Non-teaching	1.09 (0.91, 1.32)
Urban Teaching **	1.30 (1.08, 1.56)
**Hospital Region**	
Northeast	Reference
Midwest ***	0.76 (0.67, 0.86)
South	0.90 (0.80, 1.02)
West	0.89 (0.79, 1.02)

* *p* < 0.05, ** *p* < 0.01, *** *p* < 0.001.

## Data Availability

Publicly available National Inpatient Sample dataset was analyzed in this study. This data can be found from the Agency for Healthcare Research and Quality Online Healthcare Cost and Utilization Project Central Distributor: https://www.distributor.hcup-us.ahrq.gov/.

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
