# Peer review of "Burden of Mental Illness among Primary HIV Discharges: A Retrospective Analysis of Inpatient Data"

_healthcare, 2022, doi:10.3390/healthcare10050804_

Round 1
Reviewer 1 Report
Overall the paper is well written. My minor comments/suggestions are:
Title
The word impact connotes strong effect, which is commonly used in cause-effect or comparative studies, when in fact this study measures the association between mental illness and length of stay and the discussion is merely descriptive. I would suggest using a lighter word.
Introduction
In the introductory section, provide a clear statement of the study objective that reflects the study title.
Methodology
- The presence of the mental illness is not clear in the methodology; was it detected or diagnosed during the inpatient period or only during the retrospective analysis? The severity and duration of mental illness, which could have a significant difference in the LOS, were not addressed in the current study, although the severity was mentioned in the limitation.
- Please provide detail on income. Is it based on weekly, monthly or annual income?
Results
For readers' clarity, please first present the descriptive results of LOS and the association between LOS and mental illness before presenting the multivariate results.
Conclusion
- It should include the implications of these study results for improving the quality of patient care.
Reference
- More than 13 citations are older than 7 years, please provide the latest reference.
General
- Minor typo found, please check.
Author Response
Reviewer comment: The word impact connotes strong effect, which is commonly used in cause-effect or comparative studies, when in fact this study measures the association between mental illness and length of stay and the discussion is merely descriptive. I would suggest using a lighter word.
Response: Thank you for the feedback. We have addressed this and updated the title accordingly.
Reviewer comment: In the introductory section, provide a clear statement of the study objective that reflects the study title.
Response: Thank you for the feedback. We have addressed this and updated the introduction accordingly and added objective statement that relates to the title.
Reviewer comment: The presence of the mental illness is not clear in the methodology; was it detected or diagnosed during the inpatient period or only during the retrospective analysis? The severity and duration of mental illness, which could have a significant difference in the LOS, were not addressed in the current study, although the severity was mentioned in the limitation.
Response: Thank you for your feedback. We have updated the methods section to note this is diagnosed and we further clarified/defined the secondary diagnosis as well.
Reviewer comment: Please provide detail on income. Is it based on weekly, monthly or annual income?
Response: Thank you for your feedback. We have updated this to ensure it says annual.
Reviewer comment: For readers' clarity, please first present the descriptive results of LOS and the association between LOS and mental illness before presenting the multivariate results.
Response: Thank you for the feedback. We have re-organized to present the primary variables of interest first in descriptive, bivariate, and regression tables and then the rest.
Reviewer comment: It should include the implications of these study results for improving the quality of patient care.
Response: Thank you for the feedback. We have updated the discussion and the conclusion to address quality of patient care.
Reviewer comment: More than 13 citations are older than 7 years, please provide the latest reference.
Response: Thank you for your feedback. We have updated several of the citations. In paragraph starting line 269 we discussed historical trends and as such we have left those citations; however, others have been updated. In some cases we did not exclude the original study, especially if it was one of the first to establish a hypothesis, but provided newer data to support or enhance the content.
Reviewer comment: Minor typo found, please check
Response: Thank you for your feedback. We have re-read and updated any typos and errors.
Reviewer 2 Report
Line 117 - define for reader what "secondary diagnosis" is.
Line 152 - you miss one of the most staggering numbers in your entire study - low income/higher N!
Line 241 - "often due to systemic structural barriers" is assumptive and a broad brush characterization. There are other factors, of which this may be one.
Your study delivers what the title promises. This is a good study and paper for what it claims to study, but it demonstrates that there is a broader impact of the relationship between "mental illness" and HIV folk. The power of the broader impact you demonstrate is never mentioned, and is lost in the one factor of your conclusion. This paper is publishable as it is. But it is disappointing that you took such an immensely important topic and reduced your treatment of it to 12% increase in length of stay. So what? Your conclusion says that screening and follow-up will reduce the "burden of mental illness among "discharges". How? No treatment? No coordination of care?
The other issue with which you do not deal is the term "mental illness". You do not define what you mean by this, but you do use the term consistently, so, for the purposes of what you have done, strictly speaking, it works. But is that depression? Spiritual malaise? Paranoia? Mania? Is it psychological or psychiatric? My guess is that a psychiatric illness would increase "LOS" more than psychological, and would have implication for LOS. But without knowing what the term refers to, and no thought of where interface with treatment may occur, it does not mean much.
You have practically innumerable factors mentioned in your study, which dilutes its importance. And because of that you loose the importance of the relationship between "mental illness" and HIV patients, which is not reducible to having a tendency for increasing hospital stays. And is 12% meaningful compared to other maladies?
This study works, and is publishable, just as it is. It is also a crushing disappointment because it does nothing to further the understanding of the relationship of "mental illness" to HIV, other than that it increases LOS.
Author Response
Reviewer comment: Line 117 - define for reader what "secondary diagnosis" is.
Response: Thank you, we have updated this in the methods to clarify.
Reviewer comment: Line 152 - you miss one of the most staggering numbers in your entire study - low income/higher N!
Response: Thank you for your feedback. We addressed this in line 147.
Reviewer comment: Line 241 - "often due to systemic structural barriers" is assumptive and a broad brush characterization. There are other factors, of which this may be one.
Response: Thank you for your feedback. We have updated this according to the suggestion.
Reviewer comment: Your study delivers what the title promises. This is a good study and paper for what it claims to study, but it demonstrates that there is a broader impact of the relationship between "mental illness" and HIV folk. The power of the broader impact you demonstrate is never mentioned, and is lost in the one factor of your conclusion. This paper is publishable as it is. But it is disappointing that you took such an immensely important topic and reduced your treatment of it to 12% increase in length of stay. So what? Your conclusion says that screening and follow-up will reduce the "burden of mental illness among "discharges". How? No treatment? No coordination of care?
Response: Thank you for your feedback. It has brought to our attention the importance of providing more content and as such we have updated both discussion and conclusion.
Reviewer comment: The other issue with which you do not deal is the term "mental illness". You do not define what you mean by this, but you do use the term consistently, so, for the purposes of what you have done, strictly speaking, it works. But is that depression? Spiritual malaise? Paranoia? Mania? Is it psychological or psychiatric? My guess is that a psychiatric illness would increase "LOS" more than psychological, and would have implication for LOS. But without knowing what the term refers to, and no thought of where interface with treatment may occur, it does not mean much.
Response. Thank you for the feedback. We have noted in the methods that mental illness is defined as: ICD-10-CM for mood disorders (657), personality disorders (658), schizophrenia (659), suicide and intentional self-inflicted injury (662), impulse control disorders NEC (656), anxiety disorders (651), and adjustment disorders (650).
Reviewer comment: You have practically innumerable factors mentioned in your study, which dilutes its importance. And because of that you loose the importance of the relationship between "mental illness" and HIV patients, which is not reducible to having a tendency for increasing hospital stays. And is 12% meaningful compared to other maladies?
Response: Thank you for your feedback. We acknowledge that at first 12% increase in LOS may not appear too strong and as such, we have provided the full regression model, that notes that even after accounting for leading causes of morbidity and mortality, mental illness still remained a significant contributor to LOS among HIV patients. We even noted that the burden of mental illness among HIV patients was comparable to HIV patients with congestive heart failure (14% increase in LOS), and significantly more when compared to other leading causes, such as chronic pulmonary disease, diabetes, anemia, etc. We have thus highlighted this in our conclusion to bring to attention the burden of this co-morbidity as compared to other maladies.
Reviewer comment: This study works, and is publishable, just as it is. It is also a crushing disappointment because it does nothing to further the understanding of the relationship of "mental illness" to HIV, other than that it increases LOS.
Response: Thank you for your feedback. To-date, no study has been published that has looked at the outcome of HIV-mental illness co-morbidity and current studies only establish the association. As such, we have highlighted this novel contribution further in the manuscript’s conclusion as well. We have also added additional content on disposition.
Reviewer 3 Report
This manuscript is clear and presented in a well-structured manner; however, the results are not novel. Also, a significant limitation is the lack of the most critical factor related to co-morbidities in PLWH, the lymphocyte CD4 counts. Considering that the outcome is the hospital length to stay, the immunosuppression grade represents a crucial variable.
Author Response
Reviewer comment: This manuscript is clear and presented in a well-structured manner; however, the results are not novel. Also, a significant limitation is the lack of the most critical factor related to co-morbidities in PLWH, the lymphocyte CD4 counts. Considering that the outcome is the hospital length to stay, the immunosuppression grade represents a crucial variable.
Response: We thank you for your feedback. We have added a comment on CD4 counts in the limitation. However, to-date, no study has been published that has looked at the outcome of HIV-mental illness co-morbidity and current studies only establish the association. As such, we have highlighted this novel contribution further in the manuscript’s conclusion as well.
Round 2
Reviewer 3 Report
The authors made a good job improving the manuscript.